# From Posttranslational Modifications to Disease Phenotype: A Substrate Selection Hypothesis in Neurodegenerative Diseases

**DOI:** 10.3390/ijms22020901

**Published:** 2021-01-18

**Authors:** Ilia V. Baskakov

**Affiliations:** 1Center for Biomedical Engineering and Technology, University of Maryland School of Medicine, Baltimore, MD 21201, USA; baskakov@som.umaryland.edu; Tel.: +1-410-706-4562; Fax: +1-410-706-8184; 2Department of Anatomy and Neurobiology, University of Maryland School of Medicine, Baltimore, MD 21201, USA

**Keywords:** prion protein, tau, α-synuclein, neurodegenerative diseases, prion disease, Alzheimer’s diseases, Parkinson’s disease, posttranslational modifications, strains, N-linked glycans, phosphorylation, ubiquitination

## Abstract

A number of neurodegenerative diseases including prion diseases, tauopathies and synucleinopathies exhibit multiple clinical phenotypes. A diversity of clinical phenotypes has been attributed to the ability of amyloidogenic proteins associated with a particular disease to acquire multiple, conformationally distinct, self-replicating states referred to as strains. Structural diversity of strains formed by tau, α-synuclein or prion proteins has been well documented. However, the question how different strains formed by the same protein elicit different clinical phenotypes remains poorly understood. The current article reviews emerging evidence suggesting that posttranslational modifications are important players in defining strain-specific structures and disease phenotypes. This article put forward a new hypothesis referred to as substrate selection hypothesis, according to which individual strains selectively recruit protein isoforms with a subset of posttranslational modifications that fit into strain-specific structures. Moreover, it is proposed that as a result of selective recruitment, strain-specific patterns of posttranslational modifications are formed, giving rise to unique disease phenotypes. Future studies should define whether cell-, region- and age-specific differences in metabolism of posttranslational modifications play a causative role in dictating strain identity and structural diversity of strains of sporadic origin.

## 1. Introduction

In recent years, prion-like spread of misfolded, self-propagating protein aggregates was observed in a number of neurodegenerative diseases, including Alzheimer’s disease (AD), Parkinson’s disease, amyloid lateral sclerosis (ALS) and others [1,2]. In a manner similar to prion strain phenomenon, conformationally distinct self-propagating states of amyloidogenic proteins or peptides were described in several neurodegenerative diseases in humans and animal models [2,3,4,5]. Remarkably, within individual group of neurodegenerative maladies such as prion diseases, tauopathies or synucleionopathies, each of which associated with aggregation of a particular protein, diverse clinical expression could be observed [6,7,8,9]. The diversity of clinical phenotypes is attributed to the ability of amyloidogenic proteins or peptides associated with a particular disease to acquire multiple, alternative, conformationally distinct, self-replicating states [10,11,12]. By an analogy to prion strains, individual self-replicating protein states are often referred to as strains [13,14]. While the structural diversity of prion strains or strain of tau or α-synuclein has been well documented [10,13,14,15,16,17,18,19], the question how different strains of one protein elicit multiple clinical phenotypes remains poorly understood [20]. Currently, the relationship between self-replication structure and CNS response remains empirical, whereas a mechanism that would describe this relationship in a predictable manner is lacking [21].

Posttranslational modifications (PTMs) are a common feature of proteins associated with neurodegenerative diseases [22,23,24,25,26,27,28,29]. There is a growing appreciation that individual strains display different subsets of PTMs [20,30], however, specific role of PMTs in shaping strain-specific structures has not been well-defined. Do PTMs dictate what strain is formed? If so, what is the relationship between PTMs, strain-specific molecular structure and clinical manifestations it elicits?

There are two opposite views of how one can envision the role of PTMs in defining strain identity and structural diversity of self-replicating states [21]. According to one view, the spectrum of strains could be limited to only those structures which are capable of recruiting polypeptide chains regardless of the nature and position of PTMs in individual polypeptides [21]. If this is the case, most polypeptide molecules would be eligible for conversion regardless of the degree and nature of their PTMs; however, strain structural diversity is expected to be limited to only a very few structures [21]. At the opposite end, one can envision that the recruitment of polypeptide substrates by a strain is highly selective [21]. Only polypeptides with a subset of PTMs which can fit into a strain-specific structure can be successfully recruited [21]. If this is the case, a greater structural diversity is expected; however, only a sub-population of substrate molecule will be eligible for conversion by each particular strain [21]. The mechanism on selective recruitment is expected to result in a strain-specific pattern of PTMs associated with each strain [21].

## 2. Prion Diseases

Prion protein or PrP^C^ is posttranslationally modified with glycophosphatidylinositol (GPI) anchor and up to two N-glycans linked to asparagine residues in the positions 180 and 196 [27,28,29,31,32,33]. The vast majority of PrP^C^ synthesized in CNS is diglycosylated (~80%), and only small fractions are monoglycosylated (15%) and unglycosylated (less than 5%) [34,35]. While only two positions are modified with N-glycans, variations in carbohydrate structure and composition give rise to more than hundreds if not thousands of PrP^C^ sialoglycoforms [36,37]. The enormous diversity in N-glycans is attributed to the differences in glycan branching patterns, differences in structure of sialic acid residues along with different types of linkages by which sialic residues attached to galactose, diversity of naturally occurring modifications of sialic acid residues, optional fucosylation occurring at several positions, and optional modification of galactose and N-acetylglucosamine with sulfate groups [33,38,39]. As a result, PrP^C^ molecules expose diverse patterns of carbohydrate epitopes on their N-glycans [21,33]. Individual N-glycans also differ with respect to their size, which ranges from 1.5 to ~3.3 kDa [37], and their net charge, which ranges from 0 to −5 and is attributed to variations in a number of negatively charged sialic acid residues [35,40].

Recent studies revealed that among hundreds of PrP^C^ sialoglycoforms expressed by a cell, individual prion strains recruited PrP^C^ molecules selectively, according to the sialylation status of their N-linked glycans [34,38] (Figure 1). In fact, prion strains exhibited a broad range of selectivity in recruiting PrP^C^ sialoglycoforms, ranging from non-selective or weakly selective to highly selective [34,38]. In weakly selective strains, the composition of sialoglycoforms within PrP^Sc^ was very similar to that of PrP^C^, i.e., they were predominantly diglycosylated and heavily sialylated (Figure 1). In contrast, highly selective strains limited recruitment of diglycosylated and highly sialylated PrP^C^ [34,38]. As a result, they were predominantly monoglycosylated, and less sialylated than the weakly selective strains (Figure 1). Remarkably, the highly selective strains could be amplified from a mixture with weakly selective strains in protein misfolding cyclic amplification (PMCA) reactions that utilized partially deglycosylated PrP^C^ as a substrate [41]. The broad range of selectivity among prion strains appears to be attributed to strain-specific variations in tertiary and quaternary structures of PrP^Sc^ [21].

The size of N-linked glycans and their net charge, which is determined by the number of negatively charged sialic acid residues per glycan, dictate the strain-specific selectivity via imposing spatial or electrostatic constraints, respectively [34,38,40]. Among the two parameters, the size and the net charge, the charge appears to be more impactful. Indeed, providing desialylated PrP^C^ instead of PrP^C^ as a substrate for the PMCA reactions abolished selective recruitment of PrP^C^ glycoforms and restored the glycoform ratios in PrP^Sc^ to the levels typical for PrP^C^ [34,40]. In addition, desialylation of PrP^C^ was found to speed up the rates of PrP^Sc^ replication in PMCA for up to 10^6^-fold [34,40]. As expected, an increase in replication rates was found to be strain-specific [34]. These results supported the view that electrostatic repulsions between sialic acid residues on the surface of PrP^Sc^ aggregates inflict major constraints that have to be accommodated by PrP^Sc^ structure.

As predicted by the substrate selection hypothesis, selective recruitment of PrP^C^ sialoglycoforms is expected to produce strain-specific patterns of carbohydrate epitopes on PrP^Sc^ surface. In support of this hypothesis, striking differences in sialylation density was observed upon staining of PrP^Sc^ plaques formed by non-selective and highly selective strains using SNA (*Sambucus nigra*) lectin that selectively binds to sialic acid residues [21]. Moreover, strain-specific differences in sialylation patterns could be also seen using two-dimensional western blotting [20,34,42]. The differences in sialylation patterns were observed clearly even for strains with very similar selectivity [20,34,42].

Do strain-specific carbohydrate epitopes on PrP^Sc^ surface dictate disease phenotype? Several potential mechanisms should be considered for examining causative relationships between the strain-specific pattern of carbohydrate epitopes on PrP^Sc^ surface and disease phenotypes [21,38]. First, carbohydrate epitopes are likely to define the range of PrP^Sc^-binding partners, in which binding characteristics are attributed to glycan recognition [21,38,43]. Second, cell- and region-specific nuances in synthesis of PrP^C^ sialoglycoforms might create preferences for replication of individual strains in a cell- and region-specific manner, explaining the tropism of individual strains toward specific cell types or brain regions along with the region-specific neuronal vulnerability [44]. Third, the innate immune system and microglia sense sialylation levels as a cue for recognizing and clearing potential pathogens [45,46]. As such, the sialylation status of PrP^Sc^ is expected to control the rate of its clearance and its fate in an organism [47,48]. Fourth, sialylation level contribute to the pI of PrP^Sc^ particles, which, with an increase in sialylation status, is shifted from highly positive to highly negative values [49]. As such, sialylation status is expected to impact solubility, conformational stability, and aggregation states of PrP^Sc^ [20].

Indeed, recent studies provided experimental evidence supporting the aforementioned mechanisms [20,40,42,44,47,48,50]. Sialylation status of PrP^Sc^ was found to be essential for its trafficking in periphery and colonization of secondary lymphoid organs; desialylaton of PrP^Sc^ resulted in its fast clearance in periphery and CNS [42,47,48]. In agreement with the idea that PrP^Sc^ sialylation contribute to region-specific vulnerability, PrP^Sc^ sialylation was found to differ in brain regions, where the least sialylated PrP^Sc^ was found in the regions with the most severe neuroinflammation [44,51]. Studies employing cultured microglia cells and purified PrP^Sc^ offered another line of evidence toward causative relationship between sialylation status and degree of neuroinflammation [50]. PrP^Sc^ was found to induce proinflammatory response via direct contact with microglia, where the degree of response was found to be determined by the degree of sialylation of PrP^Sc^ [50].

Recent studies on adaptation of synthetic hamster prion strain to mice provided direct experimental support for a new hypothesis that selective recruitment of PrP^C^ isoforms produces unique strain-specific sialoglycoform patterns along with unique disease phenotype [20]. Six serial passages undertaken for adapting the synthetic strain to a new species were accompanied by a dramatic shift in the selectivity of recruitment of PrP^C^ sialoglycoforms giving rise to PrP^Sc^ with a unique sialoglycoform composition and disease phenotype [20]. The low sialylation status of the newly emerged strain was associated with very profound proinflammatory response and a colocalization of PrP^Sc^ with microglia, the features attributed to a unique sialoglycoform composition of PrP^Sc^ [20]. The work on cross-species prion transmission suggested that a causative relationship between a PTMs, strain-specific structure and disease phenotype exists [20].

## 3. Tauopathies

Unlike PrP^C^, tau is a subject to diverse types of PTMs, which target several dozens of amino acid residues and include phosphorylation, methylation, acetylation, glycosylation, nitration, ubiquitination and sumoylation [22,23,24]. According to a traditional view, tau hyperphosphorylation is considered to be the major trigger of tau malfunction in tauopathies [52,53]. Recent studies revealed that unique tau strains are associated with individual tauopathies including AD, chronic traumatic encephalopathy (CTE), Pick’s diseases and corticobasal degeneration (CBD) [30,54,55,56,57,58]. However, the questions how distinct tau strains are formed in different tauopathies, and whether PMTs govern formation of strain-specific structures and disease phenotypes are not well understood. Answering whether disease-specific profiles of tau PTMs exist would provide an important clue for addressing the above questions.

In a manner similar to PrP^Sc^ strains, recent studies revealed that tau strains also display unique repertoire of PTMs [30,54]. In prion diseases, PrP^Sc^ strains incorporate PrP^C^ sialoglycoforms, in which N-glycans are attached to only two residues Asn-180 and/or Asn-196 [29,31]. In contrast, in tauopathies, individual tau strains recruit tau monomers with different subsets of PMTs, which are numerous, spread along tau molecule and include unique strain-specific modifications [30,54,59].

The recent data on tau strains associated with different tauopathies suggest that the hypothesis on substrate selection might well be applicable to tau too [30]. Several PTMs, including phosphorylation, acetylation and ubiquitination at the positions S305, K311, K317, K321 and S324 were found in tau filaments associated with AD [30]. All tau isoforms carrying PTMs at aforementioned positions are eligible for conversion, since these modifications decorate the surface of AD-associated fibrils in the absence of spatial or electrostatic interference (Figure 2). The region of tau encompassing residues 285–315 is tightly packed in the interior of the fibrils associated with CBD [30]. As such, the tau isoforms modified at the positions S293, S305, K311 are expected to be excluded from recruitment to filamentous structures associated with CBD (Figure 2).

Depending on the nature of PTMs, different set of rules dictates positive or negative selection of tau isoform [30]. For instance, burying acetylated lysines into fibrillar interior might help to stabilize β-sheet stacks via neutralizing positively charged lysine residues [30]. As such, tau isoforms acetylated at positions that form fibrillar interior will be preferably selected for recruitment. Moreover, ubiquitination appears to help stabilizing the inter-filament interface [30]. Therefore, tau molecules that are ubiquitinated at the positions near inter-filament interface will be preferentially recruited by strains that rely on inter-filament interactions (Figure 2). On the other hand, by an analogy to a negative selection of hypersialylated PrP^C^ by prion strains, it is proposed here that burying negatively charged phosphate groups might create spatial and electrostatic constraints. Such constraints appear to be incompatible with CBD-specific fibrillar structure [30]. Isoforms of tau with heavy phosphorylation within the regions that are buried in fibrillar interior are expected to be excluded from recruitments in a strain-specific fashion (Figure 2).

An alternative explanation for the differences in PTMs observed between individual tau strains is that PTMs occur post-conversion. However, this hypothesis does not explain modification at the sites that are not easily accessible in fibrillar structures. Nevertheless, an important difference between the two hypotheses is that according to the substrate selection hypothesis, PTMs are proposed to play a major role in defining strain-specific structures and disease phenotype. In contrast, according to the post-conversion hypothesis, PTMs would be purely incidental. It remain to be tested whether the substrate selection hypothesis, the hypothesis on post-conversion modification or both account for strain-specific differences in PMTs of tau. Because some types of modification might stabilize fibrillar structures, whereas others provide spatial or electrostatic constraints, the strain-PTMs relationship might be very complex.

Whether PTMs associated with individual tau strains play a role in defining disease-specific clinical phenotypes is not known. Differences in cell-type specificity in transmission of tau aggregates were observed between tau strains [58,60]. For instance, only progressive supranuclear palsy- and CBD-associated tau strains induced astrocytic and oligodendrocytic tau inclusions [58]. However, it remains unknown whether strain-specific PTMs define cell tropism along with other neuropathological and clinical characteristics in tauopathies.

## 4. Synucleinopathies

Synucleinopathies are a family of diseases that includes Parkinson’s disease, multiple system atrophy (MSA) and dementia with Lewy bodies [61,62]. α-synuclein is a subject of several types of PTMs at approximately three dozen of amino acid residues including phosphorylation, ubiquitination, nitration, *O*-GlcNAcylation [25,26]. The sites for phosphorylation (S87, Y125, S129, Y133, Y135) are localized predominantly in the C-terminal acidic region (residues 96–140), the sites for *O*-GlcNAcylation (T33, T44, T54, T59, T64, T72, T75, T81, S87) are in the N-terminal amphipathic repeat region (residues 1–60) and the central hydrophobic domain (residues 61–95), whereas the sites for ubiquitination (K6, K10, K12, K21, K23, K32, K34, K43, K96) are mostly within the amphipathic repeat region (Figure 3) [25,63,64].

Akin to prion diseases and tauopathies, synucleinopathies are believed to harbor distinct strains of α-synuclein that give rise to unique disease phenotypes [4,5,14,65,66]. Indeed, several α-synuclein strains isolated from humans with synucleinopathies were transmitted to transgenic mice and produced unique disease phenotypes [4,5,65]. The question whether PTMs govern formation of distinct strains of α-synuclein and disease phenotypes has not been addressed, as the studies on this subject have been limited to in vitro approaches or cultured cells [67,68]. Nevertheless, by analogy to prion diseases and tauopathies, it is plausible that PTMs impose constraints that dictate selective recruitment of differentially modified α-synuclein isoforms into self-replicating states in a strain-specific fashion.

Recent cryo-EM studies of recombinant α-synuclein fibrils revealed two alternative structures, referred to as rod and twister polymorphs, in which well-structured β-sheet rich core comprised residues 37–99 and 43–83, respectively [69,70,71]. In rod polymorph, the central hydrophobic region comprising residues 61–95 was densely packed in fibrillar interior, whereas in twisted polymorph, a small segment of this region encompassing residues 66–78 was involved in forming interface between filaments [69]. In rod and twisted polymorphs, different segments accounted for forming interface between filaments (residues 50–57 in rod and residues 66–78 in twisted polymorphs) [69]. Nevertheless, one can predict that due to structural constraints, α-synuclein isoforms carrying *O*-GlcNAcylation at T72, T75 and T81 are excluded from both rod and twisted polymorphs and, perhaps, interfere with their replication. Consistent with this prediction, α-synuclein isoforms *O*-GlcNAcylated at T75 and T81 were found to have the greatest inhibitory effects on its aggregation in vitro [68].

Recent cryo-EM studies defined the first high-resolution structure of disease-associated α-synuclein fibrils, which were isolated from brains of patients with MSA [72]. The fibrillar material from MSA individuals consisted of two different types of fibrils, each comprised of two structurally different protofibrils, referred to as PF-IA and PF-IB (within Type I) and PF-IIA and PF-IIB (within Type II fibrils) [72]. In PF-IA and PF-IIA, the structured β-core comprised residues G14-F94, whereas the core of PF-IB and PF-IIB consisted of residues K21-Q99 and G136-Q99, respectively [72]. As both Type I and Type II fibrils were asymmetrical [72], the selectivity of recruitment of α-synuclein isoforms is likely to depend not only on the fibrillar type, but also the structure of protofibrils (Figure 3). To illustrate this point, it is expected that α-synuclein isoforms ubiquitinated at residue K60 are compatible with PF-IA- and PF-IIA-specific structures but excluded from PF-IB and PF-IIB protofibrils (Figure 3). The residue K80 is exposed to a degree sufficient for this ubiquitination to be present in PF-IIA protofibrils, but not in PF-IA, PF-IB or PF-IIB protofibrils (Figure 3). Indeed, high peptide-like density was identified near the residue K80 of PF-IIA [72]. Isoforms of α-synuclein *O*-GlcNAcylated at the residue T33 are predicted to fit only into PF-IIB protofibrils, whereas isoforms *O*-GlcNAcylated at the residue T44 are likely to be compatible with PF-IA, PF-IIA and PF-IIB, but not PF-IB protofibrils (Figure 3). Interestingly, in PF-IIB protofibrils, the residues T81-A90 were found to co-exists in two different conformations referred to as PF-IIB^1^ and PF-IIB_2_ [72]. While PF-IIB^1^ appears to be compatible with *O*-GlcNAcylation at the residues T81, PF-IIB_2_ is not (Figure 3).

To summarize, the finding that fibrils associated with MSA are structurally heterogeneous and asymmetrical raises a number of questions: Should MSA-associated fibrils be considered as one strain that consists of multiple sub-strains? Should our strain classification rely on molecular structure or disease phenotype? The substrate selection hypothesis explains how PTMs define the preferences of differentially modified protein isoforms to be recruited by structurally heterogeneous protofibrils. Assuming that PTMs metabolism is regulated in a cell type-, brain region- and age-specific manner [73,74,75], do PTMs play a role in defining strain identity and heterogeneity of self-replicating states of sporadic origin?

## 5. In Vitro Amyloids Versus In Vivo Strains

Recent years solidified the view that puzzled the prion field for years, that structures of fibrils isolated from animals or humans differ from the structures of recombinant fibrils generated in vitro [30,54,55,56,57,69,70,71,72,76,77,78,79,80]. Indeed, amyloid fibrils produced in vitro using recombinant prion protein acquire in-register parallel β-sheet structure [76,77], whereas PrP^Sc^ isolated from animals was found to have 4-rung β-solenoid structure, as illustrated by recent cryo-EM studies [78]. Structures of tau fibrils isolated from multiple tauopathies also found to differ from the structures of fibrils prepared in vitro [30,54,55,56,57,79,80]. Moreover, the first well-defined structure of fibrillar α-synuclein isolated from individuals with MSA was also different from previously determined structures of recombinant fibrils of α-synuclein produced in vitro [69,70,71,72]. Why in vivo and in vitro structures are different?

The substrate selection hypothesis might help to answer the above question. Recombinant proteins (whether the prion protein, tau and α-synuclein) used for in vitro fibrillization lack PTMs that are abundant in these proteins in a cell [69,70,71,76,80,81]. In the absence of PTMs, recombinant proteins are likely to acquire fibrillar structures that are thermodynamically and kinetically preferable, but not capable of accommodating PTMs [38,82,83,84]. Under the circumstances that PTMs impose spatial or electrostatic constraints not compatible with the in vitro structures, fibrils generated in vitro do not stand a chance of replicating in vivo. Indeed, due to electrostatic repulsion between sialic acid residues, sialoglycoforms of PrP^C^ are not compatible with the in-register parallel β-sheet structure acquired by the recombinant prion proteins in vitro [21,38].

To overcome constraints imposed by PTMs in vivo, seeding by recombinant fibrils in vivo should involve significant changes in folding patterns upon transmission of in vitro-produced recombinant fibrils to animals. The replication mechanism according to which fibrils with one cross-β folding pattern seed fibrils with different cross-β folding pattern is designated as deformed templating [85,86]. Indeed, in support of the deformed templating mechanism, transmission of fibrils of recombinant prion protein to animals was found to give rise to authentic PrP^Sc^ with structure capable of recruiting diglycosylated PrP^C^ and produced prion diseases after serial transmission that involved two or three serial passages [15,16,17,87,88].

## 6. On the Origin of Strain Diversity

As much as metabolism of PTMs is specified by a brain region, cell type and age [73,74,75], constrains imposed by PTMs on self-replicating states are also expected to be cell-, region- and age-specific. One might suggest that the cell type and brain region, from which self-propagating states originate, along with an age of an individual, are important parameters dictating an identity of strains of sporadic origin. For instance, under the circumstances that astrocytes, microglia and neurons produce substantially different subsets of PTMs, these cell types might give rise to different strains. Indeed, conformational differences between glia- and neuron-trophic strains have been noted [4,58,60,89]. However, the question whether cell-specific PTMs are responsible for these difference has not been tested.

In addition to cell- and region-specific differences, the metabolism and nature of PTMs are also regulated in a species-specific fashion [20]. As such, transmission of self-propagating states between species is expected to be a subject of new sets of constraints imposed by species-specific PTMs [20]. One should expect a drift in strain-specific properties and disease phenotype upon transmission to a new species with substantially different subsets of PTMs [20].

## 7. Summary

This article summarizes recent studies that aimed at establishing a link between strain-specific structure, PTMs and disease phenotype. In prion protein, the diversity of PTMs are attributed to the differences in structure and composition of N-glycans that are attached to only two sites. In contrast, tau is a subject to a numerous, chemically diverse range of PTMs linked to several dozens of sites. Nevertheless, general principles outlining positive and negative selection of polypeptide molecules differentially modified with PMTs might exist. On one hand, protein isoforms with the PTMs that create constraints incompatible with a strain-specific structure are expected to be excluded from conversion. On the other hand, isoforms with PTMs that stabilize intermolecular interactions are expected to be selectively recruited. It is proposed that unique, strain-specific patterns of PTMs associated with individual self-propagating structures contribute to diversity of disease phenotypes. It remains to be tested whether a causative relationship between PTMs, structure and disease phenotype exists in neurodegenerative diseases beyond prion diseases. Furthermore, it remains to be established whether cell-, region- and age-specific differences in metabolism of PTMs play a causative role in dictating strain identity and structural diversity of strains of sporadic origin. Because metabolism of PTMs is regulated, in part, in a species-specific fashion, adaptation of self-propagating states of human origin to animals is expected to be a subject of new sets of constraints imposed by species-specific PTMs. As a result, one should expect a drift in strain-specific properties and disease phenotype upon transmission of strains of human origin to animal models. This might be the case even for transmission to animal models that express human variants of the prion protein, tau, α-synuclein etc.

## Figures and Tables

**Figure 1 ijms-22-00901-f001:**
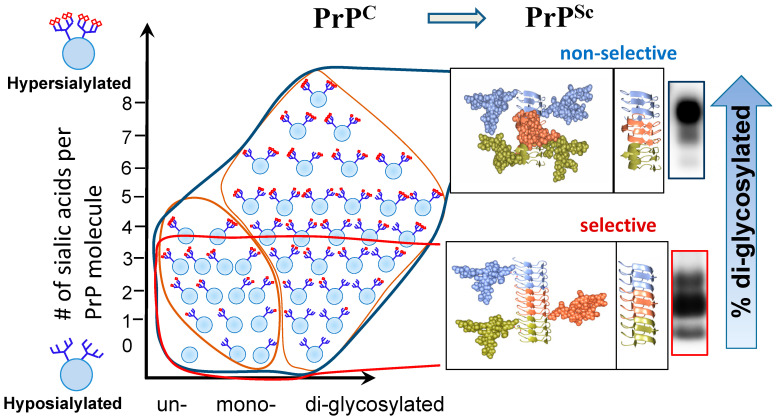
Schematic diagram illustrating selective recruitment of PrP^C^ sialoglycoforms in a strain-specific manner according to the PrP^C^ sialylation status. The left panel shows distribution of PrP^C^ molecules according to their glycosylation status (in horizontal dimension) and sialylation status (in vertical dimension) ranging from hyposialylated molecules on the bottom to hypersialylated molecules on the top. PrP^C^ molecules are shown as blue circles and sialic acid residues—as red diamonds. The panels on the right illustrating differences in quaternary assembly between non-selective (**top**) and highly selective (**bottom**) strains. While non-selective strain recruits PrP^C^ sialoglycoforms without preferences, diglycosylated and hypersialylated PrP^C^ molecules are preferentially excluded from the highly selective strains. In a non-selective strain, rotation between neighboring PrP molecules allows spatial separation of N-glycans and reduces electrostatic repulsion. In a highly selective strain, the rotation is lacking between neighboring PrP molecules. Negative selection of diglycosylated molecules helps to minimize spatial and electrostatic interference between N-glycans. As a result of selective recruitment, the ratios of glycoforms within PrP^Sc^ shift toward mono- and unglycosylated glycoform, as illustrated by corresponding western blots on the right. Adapted from [38].

**Figure 2 ijms-22-00901-f002:**
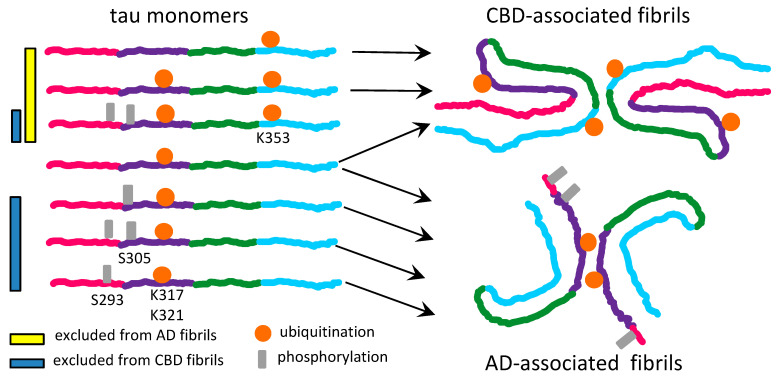
Schematic diagram illustrating selective recruitment or exclusion of differentially modified tau monomers by CBD- and AD-associated fibrils. Tau molecules phosphorylated at the residues S293 or S305 are compatible with AD- but not CBD-associated fibrils. Tau molecules ubiquitinated at the residue K353 help to stabilize CBD-associated fibrils, but appear to be incompatible with the AD-associated fibrillar structure. Tau molecules ubiquitinated at the residues K317 and K321 help to stabilize AD-associated fibrils and are compatible with the CBD-associated fibrillar structure.

**Figure 3 ijms-22-00901-f003:**
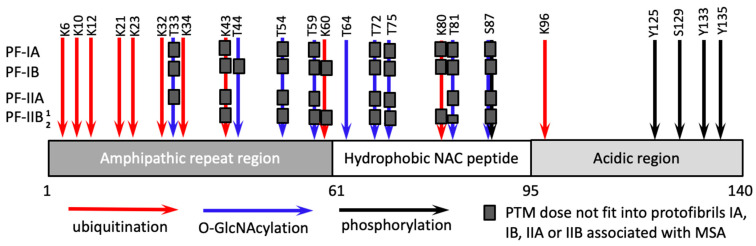
Schematic diagram illustrating site-specific PTMs, including ubiqutination, O-GlcNAcylation and phosphorylation, associated with α-synuclein. Site-specific PTMs that are not compatible with structures of PF-IA, PF-IB, PF-IIA or PF-IIB protofibrils, which constitute MSA-associated fibrils, are marked by gray boxes. Two conformations of PF-IIB protofibrils are designated as PF-IIB^1^ and PF-IIB_2_.

## Data Availability

Not Applicable.

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
