# Peer review of "From Posttranslational Modifications to Disease Phenotype: A Substrate Selection Hypothesis in Neurodegenerative Diseases"

_ijms, 2021, doi:10.3390/ijms22020901_

Round 1
Reviewer 1 Report
The review by Dr. Ilia Baskakov summarizes a body of recent findings, which emerged mostly from groundbreaking work by his team on the role of sialic acid modifications on the manifestation of strains observed in prion diseases.
In the manuscript Dr. Baskakov lays out the main findings supporting his model and extends it to other neurodegenerative diseases, specifically tauopathies and synucleinopathies.
As expected for manuscripts from an author, who enjoys a strong track record, the review is informative, thought-provoking and makes for an interesting read. In particular, the idea to extend the well-documented significance of PTMs for PrPSc strain phenomena to other neurodegenerative diseases will be noted with interest by the neudodegenerative disease research community.
Because of its novelty, this reviewer encourages Ilia Baskakov to further develop this aspect of the review by considering the following changes and additions:
- Figure 2: The ubiquitination of tau is shown as a dot in Figure 2, when in reality this modification encompasses a 76-amino-acid derivatization even in monoubiquitinated states. Although this does not invalidate the proposed role of PTMs, including ubiquitination, for strain phenomena, this reviewer suggests to focus on the role of acetylation and phosphorylation because their influence on known tau amyloid structures can be easier appreciated and is more obviously analogous to the effect sialic acid modifications have on prion strains.
- The review would also be strengthened if a cartoon was included that depicts alpha-synuclein and its main PTMs and how they might influence the formation of known strains of this protein.
- Generally, given the growing literature on tau and synuclein strain phenomena, it seems to me that the main hypothesis could be further developed by accounting more fully for reports that link PTMs to amyloid structures of these proteins
Finally, although the review is mostly well written, as is often the case with single-author manuscripts, there are numerous small errors in syntax and punctuation throughtout the text that would benefit from having another set of eyes edit the text. A few examples of errors spotted in a relatively short section of the manuscript are shown here:
Line 217: Correct "Individua" to "Individual"
Line 220: Correct "differences" to singular "difference"
Line 236: Correct "Synucleinopathies is" to "Sinucleinopathies are"
Line 269: Correct "PMTs" to "PTMs"
Line 275: Correct "This articles summarize resent" to "This article summarizes recent"
In summary, the central hypothesis underpinning this review is well-presented and argued in this manuscript. It is to be expected that this review will be of interest to a wide readership. The manuscript could be further strengthened by addressing the above points.
Author Response
Comment 1. Figure 2: The ubiquitination of tau is shown as a dot in Figure 2, when in reality this modification encompasses a 76-amino-acid derivatization even in monoubiquitinated states. Although this does not invalidate the proposed role of PTMs, including ubiquitination, for strain phenomena, this reviewer suggests to focus on the role of acetylation and phosphorylation because their influence on known tau amyloid structures can be easier appreciated and is more obviously analogous to the effect sialic acid modifications have on prion strains.
Response: We thank reviewer for this valuable suggestion. Acetylation is one of the smallest PTMs that could be easily accommodated within small fibrillar cavities. Moreover, both acetyl- and phosphate groups can stabilize fibrillar structure (as have been shown for tau and synuclein fibrils) via electrostatic interactions with positively charged amino acid side chains. For accurately predicting the site-specific effects of acetyl- and phosphate groups, whether positive or negative, one should use fibrillar atomic coordinate and specialized software. This task would be suitable for a separate research project, but is outside of the scope of the current manuscript.
Comment 2. The review would also be strengthened if a cartoon was included that depicts alpha-synuclein and its main PTMs and how they might influence the formation of known strains of this protein.
Response: New figure 3 is now included to illustrate the PTMs of a-synuclein. It depicts the effects of PTMs on formation of synuclein fibrils associated with multiple system atrophy. The first structure of synuclein fibrils associated with actual synucleinopathy has been published recently and we discuss it in great details in new paragraphs.
Comment 3. Generally, given the growing literature on tau and synuclein strain phenomena, it seems to me that the main hypothesis could be further developed by accounting more fully for reports that link PTMs to amyloid structures of these proteins.
Response: The focus of this review is on understanding the links between PTM and structures of disease-associated fibrils (i.e. fibrils originative from animal or human brains). There is a growing concerns that amyloid fibrils (of prions and now tau and synuclein) generated in vitro are not pathologically relevant ad can teach us little about these diseases. One of the question, which is not well understood and currently raised in the literature, is why amyloids produced in vitro are structurally different from in vivo strains. The revised manuscript now include a new sub-chapter that raises this topic from the perspectives of PTMs.
Comment 4. Finally, although the review is mostly well written, as is often the case with single-author manuscripts, there are numerous small errors in syntax and punctuation throughout the text that would benefit from having another set of eyes edit the text. A few examples of errors spotted in a relatively short section of the manuscript are shown here:
Line 217: Correct "Individua" to "Individual"
Line 220: Correct "differences" to singular "difference"
Line 236: Correct "Synucleinopathies is" to "Sinucleinopathies are"
Line 269: Correct "PMTs" to "PTMs"
Line 275: Correct "This articles summarize resent" to "This article summarizes recent"
Response: The text has been proof-read.
Reviewer 2 Report
From the literature, the structural diversity of strains formed by tau, α-synuclein or prion proteins has been well documented. Meanwhile, the question how different strains formed by the same protein elicit different clinical phenotypes remains poorly understood. This review shows the evidence that posttranslational modifications are important players in defining strain-specific structures and disease phenotypes and put forward a new hypothesis referred to as substrate selection hypothesis, according to which individual strains selectively recruit protein isoforms with a subset of posttranslational modifications that fit into strain-specific structures.
This review is interesting and includes very important new facts.
Unfortunately, this manuscript needs improvement and correction before publishing may be possible.
Special points
Introduction
Lines 34-37: please add references at the end of this sentence.
Lines 37-39: please add references at the end of this sentence.
Lines 39-40: please add references at the end of this sentence.
Lines 40-43: please add references at the end of this sentence.
Lines 43-45: please add references at the end of this sentence.
Lines 46-47: please add references at the end of this sentence.
Lines 47-49: please add references at the end of this sentence.
Lines 53-55: please add references at the end of this sentence.
Lines 58-59: please add references at the end of this sentence.
Lines 59-62: please add references at the end of this sentence.
Lines 63-64: please add references at the end of this sentence.
Prion diseases
Lines Lines 68-69: please add references at the end of this sentence.
Lines 87-88: please add references at the end of this sentence.
Lines 112-114: please add references at the end of this sentence.
Lines 132-134: please add references at the end of this sentence.
Lines 139-141: please add references at the end of this sentence.
Lines 146-147: please add references at the end of this sentence.
Lines 160-163: please add references at the end of this sentence.
Taupathien
Lines 181-183: please add references at the end of this sentence.
Lines 186-187: please add references at the end of this sentence.
Lines 196-197: please add references at the end of this sentence.
Lines 197-198: please add references at the end of this sentence.
Lines 203-204: please add references at the end of this sentence.
Lines 217-218: please add references at the end of this sentence.
Lines 220-221: please add references at the end of this sentence.
Lines 221-223: please add references at the end of this sentence.
Lines 223-225: please add references at the end of this sentence.
Lines 229-230: please add references at the end of this sentence.
Synucleinopathies
Lines 236-237: please add references at the end of this sentence.
Lines 239-244: please add more references at the end of this sentence.
Lines 262-265: please add references at the end of this sentence.
Summary
Please add more perspectives to your Summary part.
Author Response
Comment 1 asked to add references.
Response: Numerous new citations are provided in support of the statements that refer to previously published work. For obvious reasons, we cannot provide citations to the statements that state original ideas, thoughts or questions.
Comment 2: Please add more perspectives to your Summary part.
Response: The Summary is revised and more perspectives added.
Reviewer 3 Report
The review article “From posttranslational modifications to disease phenotype: a substrate selection hypothesis in neurodegenerative diseases” by Baskakov is a study which summarizes recent findings that are trying to establishing a link between strain-specific structure, posttranslational modifications and disease phenotype. In particular, the author puts forward a new hypothesis referred to as substrate selection hypothesis. In addition, the author also proposes that strain-specific patterns of posttranslational modifications are formed on a surface of protein aggregates which give rise to unique disease phenotypes. This hypothesis is crucial to understand pathophysiological mechanisms underlying neurodegenerative diseases including prion diseases, tauopathies and synucleinopathies. The manuscript has been well written and should be published.
Author Response
We thank this reviewer for his/her positive comments.
Round 2
Reviewer 2 Report
Thank you, now is this manuscript corrected according all my proposals.